# Hypercoagulability Evaluation in Antiphospholipid Syndrome without Anticoagulation Treatment with Thrombin Generation Assay: A Preliminary Study

**DOI:** 10.3390/jcm10122728

**Published:** 2021-06-21

**Authors:** Paul Billoir, Sébastien Miranda, Herve Levesque, Ygal Benhamou, Véronique Le Cam Duchez

**Affiliations:** 1Vascular Hemostasis Unit, Rouen University Hospital, Normandie University, UNIROUEN, INSERM U1096, F 76000 Rouen, France; veronique.le-cam@chu-rouen.fr; 2Vascular and Thrombosis Unit, Department of Internal Medicine, Rouen University Hospital, Normandie University, UNIROUEN, INSERM U1096, F 76000 Rouen, France; sebastien.miranda@chu-rouen.fr (S.M.); herve.levesque@chu-rouen.fr (H.L.); ygal.benhamou@chu-rouen.fr (Y.B.)

**Keywords:** antiphospholipid syndrome, thrombin generation assay, hypercoagulability, activated protein C resistance

## Abstract

Antiphospholipid syndrome (APS) is associated with thrombotic events (tAPS) and/or obstetrical morbidity (oAPS), with persisting antiphospholipid antibodies (aPL). Despite an update of aPL in 2006, several patients had typical clinical events without the classical biological criteria. The aim of our study was to evaluate the hypercoagulability state with both thrombin generation (TG) profiles and activated protein C resistance (aPCR) in different types of APS. Methods: We retrospectively included 41 patients with Sydney criteria classification (tAPS, oAPS) and no clinical manifestation of APS with persistent aPL (biological APS). A thrombin generation assay was performed with a Fluoroskan Ascent fluorometer in platelet-poor plasma (PPP). Activated protein C resistance was measured as a ratio: ETP_+aPC_/ETP_-aPC_ × 100. Results: Thrombotic APS and oAPS had an increase of global thrombin generation (ETP_control_ = 808 nM.min (756–853) vs. 1265 nM.min (956–1741) and 1863 nM.min (1434–2080), respectively) (Peak_control_ = 78 nM (74–86) vs. 153 nM (109–215) and 254 nM.min (232–289), respectively). Biological APS had only a lag time increase (T_control_ = 4.89 ± 1.65 min vs. 13.6 ± 3.9 min). An increased aPCR was observed in tAPS (52.7 ± 16.4%), oAPS (64.1 ± 14.6%) as compared to the control group (27.2 ± 13.8%). Conclusion: Our data suggest an increase of thrombin generation in thrombotic and obstetrical APS and no hypercoagulable states in patients with biological APS. The study of a prospective and a larger controlled cohort could determine the TGA useful for APS monitoring and could confirm an aPCR evaluation in PPP.

## 1. Introduction

Antiphospholipid syndrome (APS) is associated with thrombotic events (tAPS) and/or obstetrical morbidity (oAPS), with persisting antiphospholipid antibodies (aPL) [1]. In tAPS, a thrombotic recurrence is observed in 16% of patients. Because of anticoagulant treatments, bleeding could also occur in APS. Then mortality is about 10% at 10 years [2]. APS could be primary or secondary to autoimmune diseases (principally systemic lupus erythematosus (SLE)) [3]. However, persisting aPL in autoimmune disease is not always associated with vascular injury and thrombotic events [4]. In addition, recently, a retrospective study demonstrated a thrombotic risk increased in oAPS [5].

Moreover, several patients, were considered as “asymptomatic” APS or biological APS (bAPS), with persisting aPL without clinical event. These patients had a global thrombotic risk equivalent to that of the general population [6]. However, no biomarkers have been developed to evaluate the thrombotic risk of bAPS and possibly, to initiate primary thrombosis prevention. In humans, the hypercoagulability state can be evaluated with a thrombin generation assay (TGA) [7,8,9]. Antiphospholipid syndrome has already been evaluated by TGA. An increased lag time was observed in platelet-poor plasma (PPP) [10] and an activated protein C resistance (aPCR) in platelet-rich plasma (PRP) [10,11]. Moreover, aPL induced a hypercoagulability on endothelial cells demonstrated with TGA [12,13,14].

The aim of our study was to evaluate the hypercoagulability state in APS after a short discontinuation of anticoagulant with both thrombin generation (TG) profiles and activated protein C resistance (aPCR) in different types of APS.

## 2. Materials and Methods

We retrospectively included patients with Sydney criteria classification [15] (tAPS, oAPS) and no full criteria classification (bAPS)). Clinical data, biological results, treatment and clinical outcomes were retrieved from hospital medical records. tAPS patients who had transitory stop anticoagulant treatment were also included. Patients were considered as bAPS when an aPL positive test persisted for more than 12 weeks without clinical event. Antiphospholipid syndrome patients with congenital thrombophilia (antithrombin, protein C and S decrease, factor II and V Leiden mutation), inducing increased thrombin generation, were excluded. Patients with anticoagulant treatment, acute inflammatory syndrome, active neoplasm, or were pregnant, were excluded (TG modification). We compared to a control group of 25 healthy volunteers, without history of thrombosis.

The two different coagulation tests used to detect the lupus anticoagulant according to the International Society of Thrombosis and Hemostasis recommendations [16] were a sensitive activated partial thromboplastin time assay (PTT-LA, Diagnostica Stago, Asnières sur Seine, France) and a phospholipid dependence assay with diluted Russell viper venom test (STA-staclot DRVV screen and STAstaclot DRVV confirm, Diagnostica Stago, Asnières sur Seine, France).

Anti-cardiolipin (aCL) and anti-b2GP1 antibodies of IgM and IgG isotypes were detected in the serum by the routinely used ELISA methods: QUANTA lite IgM ACA III (Inova, WERFEN, Le Pré Saint Gervais, France) and QUANTA lite IgG ACA III (Inova, WERFEN, Le Pré Saint Gervais, France) for aCL IgM and IgG, respectively, and EliA b2-Glycoprotein I IgM (Phadia, Thermo scientific, Villebon sur Yvette, France) and EliA b2- Glycoprotein I IgG (Phadia, Thermo scientific, Villebon sur Yvette, France) for anti-b2GP1 IgM and IgG, respectively.

Thrombin generation assay was performed with Fluoroskan Ascent fluorometer (Thermoscientific Labsystems, Helsinki, Finland). Thrombin generation (TG) was triggered by a low concentration of TF (1 pM) (PPP Low reagent, Diagnostica Stago, Asnières, France). For aPCR, 6.7 nM of activated protein C were added (gift from Dr V Régnault, Université de Lorraine, INSERM, DCAC, F-54000 Nancy, France). Activated protein C resistance was measured as a ratio: ETP_+aPC_/ETP_−aPC_ × 100. All data were expressed as mean ± SD or median ± interquartile range. Data were analyzed with Graphpad Prism 5.0 (Graphpad Software Inc, San Diego, CA, United States) and with Excel 2016 (Microsoft, Albuquerque, NM, United States). One way ANOVA, Kruskall−Wallis test, and Dunn’s multiple comparison test were used. *p*-values < 0.05 were considered significant.

## 3. Results

Forty-one patients were included: 19 tAPS, 11 oAPS, 11 bAPS (5 of them with secondary bAPS). The details are reported in Table 1.

Among tAPS, ten (53%) patients had venous thrombosis, seven arterial thrombosis (37%) and two had both. Three patients had secondary APS associated to SLE. The thrombotic events occurred 5.0 ± 4.6 years before TGA. During the study, the patients were treated with antiplatelet agents (5 low dose aspirin and 2 clopidogrel). Anticoagulant treatment was resumed after the study in the tAPS group (8 VKA and 2 rivaroxaban). Four patients with SLE were treated with hydroxychloroquine (HCQ). No patients developed thrombotic events during the stop and restart anticoagulation treatment.

Among oAPS, four (36%) had early pregnancy losses, four late miscarriages (36%), two HELLP syndromes, one feature of placental insufficiency with intrauterine growth retardation. No oAPS had a thrombosis history or other systemic diseases.

Among the 11 patients identified as bAPS, 6 patients had primary bAPS, diagnosed with increased activated partial thrombin time. Five (45%) were secondary to SLE in four cases and one to rheumatoid arthritis in four cases. No bAPS patient had double or triple positivity. Five patients had a lupus anticoagulant.

### Hypercoagulability in Clinical Antiphospholipid Syndrome

Thrombin generation profiles showed a significant difference between APS groups (Figure 1A,B). When the APS groups were compared to the control group, bAPS had an increased lag time (T_control_ = 4.89 ± 1.65 min vs. 13.6 ± 3.9 min) but no difference in thrombin formation. Thrombotic APS and oAPS had an increase of global thrombin generation (ETP_control_ = 808 nM.min (756–853) vs. 1265 nM.min (956–1741) and 1863 nM.min (1434–2080), respectively) (Peak_control_ = 78 nM (74–86) vs. 153 nM (109–215) and 254 nM.min (232–289), respectively.

A nonsignificant ETP increase was observed in tAPS (*p* = 0.08) and a significant increase in oAPS (*p* = 0.02), compared to bAPS. A significant thrombin peak increase was observed in tAPS (*p* < 0.05), oAPS (*p* = 0.001), compared to bAPS. No association was observed between thrombin generation and the type or number of aPL in tAPS and oAPS. One tAPS developed a recurrent thrombosis two years after TGA (ETP: 2538 nM.min; peak: 333 nM). Two oAPS patients developed a first thrombosis during three years after test (ETP: 2067 nM.min and 1867 nM.min; peak: 254.4 nM and 232 nM).

The activated protein C resistance was evaluated in the Sydney criteria APS group. An increased aPCR was observed in tAPS (52.7 ± 16.4%), oAPS (64.1 ± 14.6%) as compared to the control group (27.2 ± 13.8%) (Figure 1C).

## 4. Discussion

In this study, we showed a TG increase in tAPS and oAPS (consistent with the Sydney criteria for diagnosis of APS), compared to the control and bAPL. The assessment of TG in a patient with tAPS was safe in our study after transient discontinuation of anticoagulant.

Moreover, TG seems to be higher in oAPS, compared to tAPS. The first thrombotic event risk has been demonstrated in oAPS recently, in a retrospective study with 63% of deep vein thrombosis development at 7.6 years after postpartum [5]. Our results support the literature data, with a hypercoagulability increase in oAPS and a susceptibility risk to develop thrombo-embolic disease.

Primary prevention in APS is still controversial. The absolute risk of a first thrombosis is probably less than 1% per year, without associated comorbidity [17], and 5.3% if a triple positivity is associated, with male sex and additional risk factors for deep vein thrombosis [6]. The results of our study suggest that bAPS did not have a hypercoagulable state with an equivalent TG to the control group. Only an increased lag time was observed, as the same mechanism of delayed activated partial thrombin time. Thrombin generation could be used to confirm no prothrombotic states in bAPS.

In a secondary bAPS without thrombotic event, only a longer lag time was observed, whereas Pereira et al. showed a TG increase in SLE [18]. In their study, six patients had secondary APS. Profiles of TG were the same between SLE and secondary APS. However, Pereira et al. evaluated the TG secondary to pro-coagulant microparticles. Our results showed a nonsignificant increase of TG in secondary bAPS. Our SLE patients were treated with HCQ. Rand et al. showed a tissue factor decrease in APS patients under HCQ [19]. An endothelial cell model of APS showed a hypercoagulable state with TG increase [12]. Preliminary results showed that the prothrombotic state decreased when a mouse and endothelial cell model of APS were treated with HCQ [13]. The hydroxychloroquine treatment could explain the absence of difference between secondary bAPS and the control group in thrombin generation profiles.

Activated protein C resistance is known to increase venous thrombotic risk in APS with an odd ratio equal to 3.31 (1.30–8.41) [20]. This aPCR was demonstrated in platelet rich plasma with TGA [11]. We showed an aPCR in tAPS, oAPS. A study by Zuily et al. confirmed an aPCR in APS, in a univariate analysis, superficial venous thrombotic inducing an increased risk of venous thrombotic development [21]. Our results confirmed the Zuily et al. study in aPCR on APS, and the possibility of performing aPCR research in PPP, which would permit delayed analysis.

Nevertheless, our study presents some limitations. First, given the small sample size, the study was not able to perform subgroup analysis or to compare subgroups of patients according to their arterial or venous presentation. Moreover only three patients had triple positivity in the entire cohort which prevented a comparison with any other antibody profile. Thrombin generation was performed after transient discontinuation of anticoagulant treatment in tAPS.

## 5. Conclusions

Our data suggest an increase of thrombin generation in thrombotic and obstetrical APS which was not found in patients with bAPS. The thrombin generation profile appears to be a reliable marker of hypercoagulability in APS after a short discontinuation of anticoagulant. A prospective follow up of a larger APS cohort might help to determine individual risk for a first thrombotic event in bAPS and oAPS, as well as the risk of recurrent thrombotic events in APS.

## Figures and Tables

**Figure 1 jcm-10-02728-f001:**
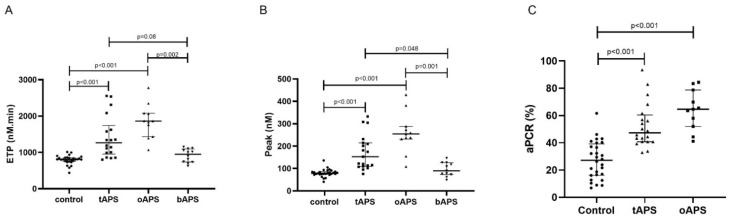
Thrombin generation profiles in antiphospholipid syndrome patients. (**A**) Endogenous thrombin potential. (**B**) Thrombin peak. (**C**) Activated protein C resistance. Kruskall−Wallis and Dunn’s multiple comparison test were used. tAPS: thrombotic antiphospholipid syndrome. oAPS: obstetrical antiphospholipid syndrome. bAPS: biological antiphospholipid syndrome. Circle is for control. Square is for thrombotic antiphospholipid syndrome in figure A et B, and obstetrical antiphospholipid for C. Triangle is for obstetrical antiphospholipid syndrome in figure A et B, and thrombotic antiphospholipid for C. tAPS and oAPS are described in legend.

**Table 1 jcm-10-02728-t001:** Clinical and biological characteristics of patients. tAPS: thrombotic antiphospholipid syndrome. oAPS: obstetrical antiphospholipid syndrome. bAPS: biological antiphospholipid syndrome. EPL: early pregnancy losses. IGR: placental insufficiency with intrauterine growth retardation. APL: antiphospholipid antibodies. LA: lupus anticoagulant. aCL: anticardiolipine. Anti-β2GPI: anti-beta-2-glycoprotein I.

	Control	tAPS	oAPS	bAPS
N	28	19	11	11
Age (years)		49.5 ± 18.3	36.3 ± 8.8	57.1 ± 19.7
Sex (women/men)		16/3	11	10/1
Thrombotic event				
Venous thrombosis		10		
Arterial thrombosis		7		
Both		2		
Obstetrical complication				
EPL			4	
Late miscarriages			4	
HELLP syndrome			2	
IGR			1	
APL				
LA		4	2	8
aCL IgM/IgG		7/5	1/6	5/3
Anti-β2GPI IgM/IgG		0/4	1/1	3/0
Double positivity		1	1	
Triple positivity		2	2	3

## Data Availability

The data are available from the corresponding author (paul.billoir@chu-rouen.fr).

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
