# Peer review of "Hypercoagulability Evaluation in Antiphospholipid Syndrome without Anticoagulation Treatment with Thrombin Generation Assay: A Preliminary Study"

_jcm, 2021, doi:10.3390/jcm10122728_

Round 1

Reviewer 1 Report

In this article entitled ”Hypercoagulability evaluation in antiphospholipid syndrome without anticoagulation treatment with thrombin generation assay: a preliminary study” the authors focus on the increase of thrombin generation in thrombotic and obstetrical APS and no hypercoagulable states in patients with biological APS. The present article is engaging, and is of potential interest to the readers of Journal of Clinical Medicine.

I recommend the publication of this work only after the authors have addressed the following major comment:

-To improve the evaluation of the hypercoagulability state the authors should provide the plasma levels of TF released in the different types APS.

-Minor comment:

On page 4 lines 136-137, I think this sentence should be eliminated.

Author Response

Reviewer 1.

Comments and Suggestions for Authors

In this article entitled ”Hypercoagulability evaluation in antiphospholipid syndrome without anticoagulation treatment with thrombin generation assay: a preliminary study” the authors focus on the increase of thrombin generation in thrombotic and obstetrical APS and no hypercoagulable states in patients with biological APS. The present article is engaging, and is of potential interest to the readers of Journal of Clinical Medicine.

I recommend the publication of this work only after the authors have addressed the following major comment:

-To improve the evaluation of the hypercoagulability state the authors should provide the plasma levels of TF released in the different types APS.

We understand the reviewer suggestion.

Activation of endothelial cells and monocytes by anti-phospholipid antibodies results in over-expression of TF and adhesion molecules, which are considered to be major mechanisms leading to thrombosis in APS (10.1017/S1462399407000506). In obstetrical APS, increase of TF was described in mouse model (10.1097/OGX.0b013e3181c97809).

We did not initially measure tissue factor in APS patients. If the editor grants us additional time for revision (till 06/16/21), we can perform analysis.

-Minor comment:

On page 4 lines 136-137, I think this sentence should be eliminated.

We thank the reviewer for this suggestion.

We have performed the modification.

Reviewer 2 Report

In this interesting paper  Bolloir et al uses thrombon generation assays to evaluate the thrombotic potential in APLA patients ( thrombotic , obstetric and asymptomatic).

The number of patients in each group is small, especially the bAPLA triple positive group but the results are important. 

I have a suggestion regarding thr APCR results. since factor V Leiden mutation might effect this test it is important to provide information about the mutational status of the patients especialy among the tAPLA group.

In the tAPLA group it is important to mention if othr abnormalities were found in their thrombophilia workup.  

Author Response

Reviewer 2.

Comments and Suggestions for Authors

In this interesting paper  Bolloir et al uses thrombon generation assays to evaluate the thrombotic potential in APLA patients ( thrombotic , obstetric and asymptomatic).

The number of patients in each group is small, especially the bAPLA triple positive group but the results are important. 

I have a suggestion regarding thr APCR results. since factor V Leiden mutation might effect this test it is important to provide information about the mutational status of the patients especialy among the tAPLA group.

In the tAPLA group it is important to mention if othr abnormalities were found in their thrombophilia workup.  

We understand reviewer comment.

We performed in our antiphospholipid syndrome patient congenital thrombophilia research (antithrombin, protein C and S decrease, factor II and V Leiden mutation). None had congenital thrombophilia.

We have added in material section:

‘’Antiphospholipid syndrome patient with congenital thrombophilia (antithrombin, protein C and S decrease, factor II and V Leiden mutation), inducing increase thrombin generation, has been excluded.’’

Round 2

Reviewer 1 Report

Accept in present form